# Kathon Induces Fibrotic Inflammation in Lungs: The First Animal Study Revealing a Causal Relationship between Humidifier Disinfectant Exposure and Eosinophil and Th2-Mediated Fibrosis Induction

**DOI:** 10.3390/molecules25204684

**Published:** 2020-10-14

**Authors:** Mi-Kyung Song, Dong Im Kim, Kyuhong Lee

**Affiliations:** 1National Center for Efficacy Evaluation of Respiratory Disease Product, Korea Institute of Toxicology, 30 Baehak1-gil, Jongeup, Jeollabuk-do 56212, Korea; mikyung.song@kitox.re.kr (M.-K.S.); dongim.kim@kitox.re.kr (D.I.K.); 2Department of Human and Environmental Toxicology, University of Science and Technology, Daejeon 34113, Korea

**Keywords:** humidifier disinfectant, Kathon, chloromethylisothiazolinone, methylisothiazolinone, fibrosis, Th2

## Abstract

Currently available toxicity data on humidifier disinfectants are primarily limited to polyhexamethylene guanidine phosphate-induced lung fibrosis. We, therefore, investigated whether the sterilizer component Kathon, which is a mixture of chloromethylisothiazolinone and methylisothiazolinone, induces fibrotic lung injury following direct lung exposure in an animal model. Mice were intratracheally instilled with either the vehicle or Kathon. Differential cell counts, cytokine analysis, and histological analysis of lung tissue were then performed to characterize the injury features, and we investigated whether Kathon altered fibrosis-related gene expression in lung tissues via RNA-Seq and bioinformatics. Cell counting showed that Kathon exposure increased the proportion of macrophages, eosinophils, and neutrophils. Moreover, T helper 2 (Th2) cytokine levels in the bronchoalveolar lavage were significantly increased in the Kathon groups. Histopathological analysis revealed increased perivascular/alveolar inflammation, eosinophilic cells, mucous cell hyperplasia, and pulmonary fibrosis following Kathon exposure. Additionally, Kathon exposure modulated the expression of genes related to fibrotic inflammation, including the phosphatidylinositol 3-kinase (PI3K)/protein kinase B (AKT) signaling pathway, extracellular signal regulated kinase (ERK)1 and ERK2 cascade, extracellular matrix (ECM)-receptor interaction pathway, transforming growth factor beta receptor signaling pathway, cellular response to tumor necrosis factor, and collagen fibril organization. Our results suggest that Kathon exposure is associated with fibrotic lung injury via a Th2-dependent pathway and is thus a possible risk factor for fibrosis.

## 1. Introduction

Exposure to humidifier disinfectants (HDs) has been identified as a potential cause of an outbreak of lung injury in South Korea [1]. Over the past seventeen years, various types of HD have been used, to which a large proportion of the Korean population has been exposed. The magnitude of the damage caused by these HDs is postulated to be considerably larger than previously demonstrated [2].

Isothiazolinones are used in cosmetic and chemical substances for occupational use. Isothiazolinones are also multi-purpose biocides that are widely used in consumer products, as well as in HDs for their sterilizing properties [3]. The most frequently used isothiazolinones are 5-chloro-2-methyl-4-isothiazolin-3-one (CMIT) and 2-methyl-4-isothiazolin-3-one (MIT), which are generally applied at a CMIT:MIT ratio of 3:1 [4]. The major chemical disinfectants included in HDs are a mixture of CMIT/MIT and other chemical compounds, including polyhexamethylene guanidine phosphate (PHMG-P) and oligo (2-(2-ethoxy)ethoxyethyl) guanidine chloride (PGH). Previous studies have reported that PHMG-P and PGH can cause lung injury, defined as HD-associated lung injuries (HDLIs) [5,6,7]. Moreover, a previous case report investigating HDLI pathology reported that CMIT/MIT can also induce similar lung injury pathology as that caused by PHMG/PGH [8]. This effect has been clinically confirmed in several human cases in which HD products containing a mixture of CMIT/MIT have been used in the home [9]. Further, a recent study demonstrated that Kathon may induce eosinophilia-mediated lung disease [10] and cause atopic dermatitis symptoms via dysregulation of Th2/Th17-related immune responses [11].

Despite these findings, previous animal studies have not conclusively characterized the causal relationship between HDs containing CMIT and/or MIT and fibrotic lung injury. Therefore, in this study, we investigated whether a HD containing CMIT and MIT induces fibrotic lung injury following direct lung exposure in a mouse model.

## 2. Results

### 2.1. Changes in Organ Weight

Relative lung weight was significantly increased in groups that received 1.14 mg/kg Kathon (*p* < 0.0001), and no significant changes in weight was observed in the spleen and thymus (Figure 1).

### 2.2. Histological Changes in Lung Samples

Hematoxylin and eosin (H&E) and Masson’s trichrome (MT) staining clearly demonstrated the pathological features of inflammation and fibrosis in the Kathon-treated groups, but not in the controls. Specifically, the left lungs of the control group had a normal histological appearance, while the Kathon group lung showed extensive damage characterized by granulomatous inflammation/fibrosis and alveolar macrophage infiltration (Figure 2 and Appendix A). As assessed via MT staining, the Kathon group presented with severe fibrotic lesions and collagen deposition compared to the saline control group. The Kathon groups also showed eosinophilic responses in pulmonary vessels. Evaluation of H&E-stained lung sections revealed significantly higher perivascular eosinophilic cell infiltration scores in the Kathon groups than in the saline control group. Additionally, evaluation of periodic acid-Schiff (PAS)-stained lung sections revealed elevated goblet cell hyperplasia in the Kathon groups compared to in the control group.

### 2.3. Cellular Changes and Cytokine Levels in BALF

Examination of bronchoalveolar lavage fluid (BALF) revealed that intratracheal Kathon instillation dose-dependently increased the total number of differential cells, which was significantly increased in the 1.14 mg/kg Kathon dose group (Figure 3). A significant difference was observed in the cytological composition between the Kathon groups and the control group. The proportion of eosinophils and neutrophils in BALF increased in a dose-dependent manner in the Kathon group. The proportion of eosinophils increased in the 0.57 and 1.14 mg/kg Kathon dose groups, with eosinophils comprising approximately 16% of the total BALF cells in the 1.14 mg/kg dose group. Similarly, the proportion of neutrophils significantly increased to approximately 15% and 30% of the total BALF cells in the 0.57 and 1.14 mg/kg Kathon dose groups, respectively. The absolute number of macrophages, eosinophils, and neutrophils was also significantly increased in the Kathon groups, particularly in the 1.14 mg/kg dose group.

We assumed that eosinophils are significantly involved in Kathon-induced fibrotic inflammation. The eosinophils are key contributors to T helper 2 (Th2)-associated pathologies which play important roles in fibrotic responses during airway remodeling. Therefore, we assessed T helper 2 (Th2) cytokine levels (interleukin (IL)-4, IL-5, and IL-13) in BALF. Results show that IL-4 and IL-5 levels were significantly higher in the BALF of mice exposed to 1.14 mg/kg Kathon than in the control group (Figure 4). Moreover, IL-13 levels increased in both the 0.57 and 1.14 mg/kg Kathon dose groups, however, no significant changes were observed compared to the control group.

### 2.4. Gene Expression Changes Associated with Inflammation and Asthmatic Responses

Next, we identified the gene expression patterns altered by Kathon to reveal the molecular mechanism associated with Kathon-induced fibrotic responses. Given that fibrotic features were particularly high in the 1.14 mg/kg dose group, transcriptomic analysis was performed only in this group (only RNA samples with A260/A280 ratios >1.8 and RNA integrity number (RIN) values >7 were used; Appendix A). A combination of fold change ≥1.5 and *p* < 0.05 was used to define differentially expressed genes (DEGs). Exposure to 1.14 mg/kg Kathon lead to the up- and downregulation of 497 and 702 genes, respectively. The top ten genes with the highest fold change were *Clca1* (255.01-fold), *Chil4* (150.22-fold), *Itln1* (130.40-fold), *Tff1* (27.30-fold), *Muc5ac* (23.22-fold), *Clca3b* (23.12-fold), *Chodl* (22.06-fold), *Pla2g4c* (16.86-fold), *Mmp10* (13.45-fold), and *Stac2* (12.70-fold).

Key biological processes and pathways were also significantly affected by Kathon exposure, as revealed by expression analysis systematic explorer (EASE) analysis (Table 1). Kathon-altered genes were primarily associated with inflammatory responses and fibrotic inflammation, such as cellular response to IL-1, positive regulation of the mitogen-activated protein kinase (MAPK) cascade, cell responses to tumor necrosis factor (TNF), extracellular matrix (ECM) organization, TGF-β receptor signaling pathway, collagen fibril organization, PI3K-AKT signaling pathway, and ECM-receptor interaction.

Next, using the Comparative Toxicogenomics Database (CTD) database, genes were further analyzed to elucidate gene-disease associations, within the respiratory tract disease category, that are altered by Kathon exposure. Analysis of 1199 genes revealed that 107 were involved in various respiratory tract-related diseases (i.e., obstructive lung diseases, respiratory hypersensitivity, bronchogenic carcinoma, asthma, pulmonary fibrosis and chronic obstructive pulmonary disease), and twelve genes (*Ace2*, *Acta2*, *Adipoq*, *Areg*, *Cat*, *Cfd*, *Col3a1*, *Eln*, *Fn1*, *Il12b*, *Sod1*, and *Wnt5a*) were involved in pulmonary fibrosis (Table 2).

Here, we assumed that the Th2 pathway might be involved in Kathon-induced fibrosis. Our further analysis using ingenuity pathway analysis (IPA) shows that most of the genes involved in the Th2-related pathways, including the Th2 pathway and IL-4 signaling, were upregulated or predicted to be activated (Figure 5). In total, 15 genes (*Bmpr2*, *H2eb2*, *H2q7*, *H2aa*, *H2eb1*, *Icosl*, *Il12b*, *Il1rl1*, *Maf, Notch4*, *Runx3*, *S1pr1*, *Tgfbr1*, *Tgfbr3*, and *Akt3*) were involved in the Th2 and IL-4 signaling pathways (Table 3).

## 3. Discussion

A recent study performed by our research group showed that Kathon induces apoptotic cell death along with membrane damage in human bronchial epithelial (BEAS-2B) cells [10], while, several inflammatory responses, including total number of BALF cells and the levels of TNF-α, IL-5, IL-13, MIP-1α, and MCP-1α, were markedly increased in the lung of Institute of Cancer Research (ICR) mice after single instillation with Kathon. The proportion of natural killer cells and eosinophils were also significantly elevated in the spleen and blood, respectively. We, therefore, postulated that Kathon may induce eosinophilia-mediated disease in the lung by disrupting homeostasis of pulmonary surfactants. Considering that eosinophilia is closely related to fibrosis, in the current study we sought to investigate whether Kathon induces fibrotic lung injury following direct lung exposure and alters fibrosis-related gene expression in lung tissues, using an animal model. Since our preliminary studies found that single dose Kathon did not elicit an apparent effect on lung injury, in the current study Kathon was repeatedly administered to allow for the clear investigation of Kathon-induced lung injury.

Animals were exposed to Kathon at 0.57 and 1.14 mg/kg, with the dosage used for this investigation determined based on previous preliminary studies by our research group. The dose correlates with the actual dose to which humans are exposed via humidifier disinfectant products. The HD concentration recommended by the manufacturer was 0.23 mg/m^3^. The concentration of the active ingredient in air, considering the CMIT/MIT content in the product, is 0.014 mg/m^3^. When calculated as a dosage to be delivered to the murine model, this dose is equivalent to 0.033 mg/kg. The dose used in this study is approximately 17–35-fold higher than the recommended concentration. However, considering that the HD product was used at more than 5–10-fold the recommended concentration, the dose used in this study is not considered to be unreasonably high. Furthermore, the risk assessment for all toxic endpoints should consider differences between animal species, individual variations in sensitivity, and uncertainty factors, including limitations of the study, animal-to-human extrapolation, sensitive subpopulations, and database inadequacies. When the standard 10-fold interspecies and 10-fold intraspecies uncertainty factors are applied, the difference in sensitivity between the test species and humans is at least 100-fold. Considering these factors, the doses of Kathon used in the present in vivo study are within a sufficiently reasonable range. Therefore, we believe the doses are appropriate for analyzing the effect of Kathon in vivo and are sufficiently predictive of the effects in humans.

Results of the current study indicated that PHMG-P and Kathon, two major components of HD, induced equivalent levels of lung fibrosis. Histopathologic findings revealed typical fibrotic inflammation, similar to the PHMG-P-induced fibrotic models and diseased human lungs. However, in a previous study, significant weight loss was observed in the thymus of a PHMG-P-induced fibrosis model [12], while no thymus weight change was observed in mice exposed to Kathon in the current study. These results suggest that although Kathon may induce lung injury, it appears to do so via a mechanism that differs from that employed by PHMG-P.

Additionally, Kathon displayed discrepant cytological compositions. In the Kathon groups, there were significant increases in neutrophils and eosinophils and no changes in the lymphocytes. This is in contrast to the results for the PHMG-P-induced fibrosis model, which showed a significant increase in neutrophils and lymphocytes and a lack of significant change in eosinophils [13]. Moreover, Kathon-induced immune differential cell accumulation showed a mixed granulocytic inflammation pattern, to which macrophages, eosinophils, and neutrophils contributed. However, the increase in eosinophils was of particular interest, as this is consistent with the histological changes showing increased perivascular eosinophilic cell infiltration in the Kathon groups. We, therefore, assumed that eosinophils are involved in Kathon-induced fibrotic inflammation.

Leukocytic infiltration is a characteristic of lung fibrosis and is believed to contribute, at least in part, to the fibrotic response [14,15]. Among these recruited cells, macrophages and their profibrotic activity have been the focus of intense research. However, the precise role played by other cells has not yet been fully delineated [16,17,18,19]. This is particularly true for eosinophils, which are also recruited during the establishment of fibrosis. Eosinophils are frequently associated with tissue remodeling and fibrosis in allergic reactions, other diseases, and animal models. As a major source for several key fibrogenic cytokines during the early stages of fibrosis, eosinophils may contribute to lung injury and the development of fibrosis [20,21]. Indeed, eosinophil accumulation in the alveolar space and parenchyma has been observed in fibrogenic lesions, such as idiopathic pulmonary fibrosis in humans and bleomycin (BLM)-induced lung fibrosis in rats and mice [22,23,24]. Eosinophils are key contributors to Th2-associated pathologies, and Th2 cytokines play important roles in fibrotic responses during airway remodeling. In the current study, Th2-associated cytokines, especially IL-4 and IL-5, in BALF were significantly increased in the Kathon groups. These cytokines have been causally linked to the development of fibrosis in a variety of chronic inflammatory diseases [25]. Notably, these cytokines are strongly involved in tissue remodeling and fibrosis [26]. IL-4 and IL-13 share many biological functions; for example, they both drive the differentiation of resident and recruited fibrocytes to myofibroblasts in a range of tissues [27,28,29]. Moreover, in transgenic mice that specifically overexpress IL-4 or IL-13 in the lung, both cytokines function as profibrotic mediators by directly and indirectly influencing myofibroblast activation [30,31]. Meanwhile, IL-5 can also promote fibrosis in the lung by recruiting eosinophils. These cells are an important source of pro-fibrotic cytokines and growth factors, such as transforming growth factor (TGF)-β1 and IL-13 [32,33,34,35]. Further, IL-5 is reportedly upregulated in the eosinophils of BLM-induced fibrotic mouse lungs [24,36]. Taken together, these results suggest that Kathon-induced fibrotic responses show a Th2-related mechanism, to which eosinophils contribute.

We then postulated that specific changes in gene expression were involved in Kathon-induced Th2-related fibrosis. Therefore, we analyzed the Kathon-altered gene expression patterns to reveal the molecular mechanism associated with Kathon-induced fibrotic responses. The top ten genes (*Clca1*, *Chil4*, *Itln1*, *Tff1*, *Muc5ac*, *Clca3b*, *Chodl*, *Pla2g4c*, *Mmp10*, and *Stac2*) with the highest fold change are involved in various inflammatory responses and respiratory diseases. Recent research has recognized a connection between calcium-activated chloride channel (*Clca*) gene expression and the development of inflammatory airway disease, both in animal models and in humans with asthma, chronic obstructive pulmonary disease (COPD), and cystic fibrosis. Moreover, *hCLCA1* upregulation has been observed in the airways of cystic fibrosis patients with high levels of mucus production [37,38]. *Clca1* not only regulates mucin expression but also participates in innate immune responses by binding to yet unidentified molecules on inflammatory cells for cytokine and chemokine production. Ectopic expression of either *Clca3* or its human homolog *hCLCA1* in the human mucoepidermoid cell line NCI-H292 results in *Muc5ac* upregulation [39]. Human CLCA1 is reportedly closely associated with mucus production in the bronchial epithelium of asthma patients [40]. Moreover, increased *hCLCA1* expression has recently been reported in the nasal and sinus mucosa of cystic fibrosis patients. Stimulation with the Th2 cytokines IL-4 and IL-13 significantly increases hCLCA1 protein expression in mucosal tissue explants from the upper airways of fibrosis patients [37,41]. Therefore, similar to in asthma, *CLCA1* may contribute at least in part to mucus overproduction during fibrosis.

Chitinase-like proteins (CLP), including CHIL4, have been linked to the activation of an alternative macrophage phenotype (M2) [42,43]. This phenotype is found in asthma and other chronic diseases, such as cystic fibrosis, supporting the notion that chitinases and CLPs participate in these disease conditions. Indeed, the levels of circulating CLP correlate with subepithelial fibrosis in asthmatic airways [44], and studies demonstrate increased levels of circulating CLP in pulmonary fibrosis [45]. In accordance with these findings, recent investigations have focused on the role of CLP in fibrotic and hyperproliferative remodeling responses. These studies demonstrated that CLP plays a key role in IL-13-induced pulmonary fibrosis and TGF-β1 elaboration [43]. Additionally, previous studies have implicated *MMP10* as a novel biomarker for idiopathic pulmonary fibrosis, reflecting both disease severity and prognosis in patients with idiopathic pulmonary fibrosis [46]. These gene expression changes are consistent with cytokine analysis and histological changes observed in the current study, showing an increase in IL-4, IL-13, fibrotic inflammatory lesions, and mucus cell hyperplasia (Figure 2 and Figure 3).

Functional analysis with Kathon-altered genes revealed that a significant proportion of genes were related to the inflammatory responses and fibrotic inflammation, such as cellular response to IL-1, positive regulation of the MAPK cascade, cell responses to TNF, ECM organization, TGF-β receptor signaling pathway, collagen fibril organization, PI3K-AKT signaling pathway, and ECM-receptor interaction. TGF-β1 and TNF-α upregulation is believed to play important role in the development of pulmonary fibrosis. In the lung, TNF-α and TGF-β1 are regarded as prototypical “profibrotic” mediators, which are key cytokines in extracellular matrix production, and may increase fibroblast proliferation, differentiation, and extracellular matrix deposition, as well as promote matrix metalloproteinase (MMP) induction. These MMPs enhance basement membrane disruption and can facilitate fibroblast migration [47,48]. Moreover, evidence suggests that TNF-α influences TGF-β1 expression [49]. Previous studies have shown that an anti-TNF-α antibody reduced BLM-induced lung injury and decreased TGF-β1 expression in treated mice [50]. Likewise, TNF-α overexpression in the lungs of normal rats induced interstitial fibrosis and was preceded by a significant increase in TGF-β1 production [49]. TNF-α has also been observed to induce MAPK phosphorylation, which may contribute to transcription factor activation and mRNA stability for fibrosis-related cytokines and other growth factor genes [51]. Extracellular signal-regulated kinase (ERK), MAPK, and the phosphatidyl inositol 3-kinase (PI3K)/AKT pathway have also been shown to be involved in TGF-β1-induced fibrosis [52,53,54]. Moreover, Kathon-altered genes were also linked to neutrophil chemotaxis, a characteristic feature of pulmonary fibrosis. Changes in these genes are consistent with the observed changes in BALF cell counts, which show a significant increase in neutrophils (Figure 3). Cumulatively, these results suggest that the Kathon-altered genes and related signaling pathways may create a microenvironment conducive to the development of Kathon-related lung fibrosis.

CTD analysis revealed that 107 genes were involved in various respiratory tract-related diseases (i.e., obstructive lung diseases, respiratory hypersensitivity, bronchogenic carcinoma, asthma, pulmonary fibrosis and COPD). Many reports show that these genes play an important role in regulating inflammatory responses, consequently inducing fibrosis. Some of these genes exhibited decreased expression patterns in the lungs of Kathon-exposed mice (i.e., *Ace2*, *Acta2, Adipoq, Cat, Cfd,* and *Sod1*). Angiotensin converting enzyme-2 (ACE2) is primarily detected in the lung epithelium and vascular endothelium, however, is significantly decreased in idiopathic pulmonary fibrosis lungs. It has recently been shown that ACE2 attenuates BLM-induced lung fibrosis [55]. Previous studies proposed that ACE2 can block TGF-β1-induced epithelial–mesenchymal transition (EMT) in alveolar epithelial cells, ACE2 overexpression can inhibit TGF-β1 expression by human fibroblasts, and ACE2 can suppress cell proliferation and collagen synthesis [56,57]. Additionally, loss of superoxide dismutase 1 (SOD1) from pulmonary metrics leads to reactive oxygen species (ROS) accumulation, further promoting the inflammatory response and enhancing fibrosis pathology. Targeted SOD overexpression in the lungs of mice significantly protects mice against BLM-induced lung injury [58], while enhanced BLM-induced pulmonary damage occurs in mice lacking SOD [59]. SOD can also stabilize the ECM components by preventing ROS-induced degradation, thus preventing TGF-β activation through ECM-stimulated mechanisms [60]. These genes may be regarded as signature genes reflecting the responses to Kathon-induced lung injuries including fibrosis.

The current study has several limitations. First, intratracheal instillation is not a satisfactory alternative to inhalation and may result in markedly different distribution, transport, and toxicity of the test substances in the lungs. However, our previous study which investigated the PHMG-P-induced fibrotic effects, confirmed that intratracheally instilled animals exhibited similar fibrotic features with inhaled animals. Therefore, we propose that intratracheal instillation is an acceptable model for observing HD-induced toxic effects, however, the associated results require further validation using inhalation studies. Second, the clear molecular mechanisms by which Kathon induces fibrotic effects remain unknown. Although our data are insufficient to reflect the entire fibrotic responses induced by HDs, this is the first study to demonstrate the association of Kathon exposure with fibrotic inflammation in an animal model. Further studies on inhalation exposure are warranted to evaluate the fibrosis-inducing effects of Kathon and their potential mechanisms.

## 4. Materials and Methods

### 4.1. Animals and Exposure Protocol

Male C57BL/6 mice (Orient Bio Ltd., Seongnam, Korea) were housed in light- (12 h light/12 h dark cycle) and temperature-controlled (22 ± 3 °C) rooms. Throughout the experiments, the mice had free access to standard laboratory chow and tap water. After 10 days of acclimation, the mice used for in the study exhibited normal weight gain and no adverse clinical signs. The experiments were performed with the approval of the Institutional Animal Care and Use Committee of the Korea Institute of Toxicology (approve no. 2003-0063, date 2020.03.06).

The study consisted of six groups defined by their specific exposure profile (each group, *n* = 5): naive control, vehicle control, and two Kathon exposure groups with different doses (0.57 and 1.14 mg/kg). The mice were randomly divided into four weight-matched experimental groups using the Pristima System (Version 7.x; Xybion Medical Systems Corporation, Lawrenceville, NJ, USA).

A concentrated stock solution of Kathon (1.492% DOW Chemical Com. Midland, MI, USA) was diluted in saline to create equivalent doses of 0.57 and 1.14 mg/kg, which were instilled intratracheally using a modified automatic video instillator (Doobae System, Seoul, Korea). Kathon was administered on days 1, 4, 6, 8, 11, and 13 (Appendix A). For intratracheal instillation, mice were anesthetized using inhaled anesthesia. Prior to instillation, isoflurane was delivered into an induction chamber using small animal portable anesthesia systems (L-PAS-02, LMSKOREA Inc, Seongnam, Korea) equipped with an isoflurane vaporizer. Mice were then exposed to 2.5% isoflurane delivered in O_2_ (2 L/min) within the induction chamber until a sleep-like state was achieved. Mice that received isoflurane anesthesia were removed from the induction chamber and instillation was performed immediately.

Forty-eight hours after the final instillation with Kathon, mice were euthanized with an overdose of isoflurane and continuously exposed until one minute after breathing stopped. Samples were collected from the euthanized animals for further analysis.

### 4.2. BALF Preparation

Forty-eight hours after the final Kathon instillation, mice were anesthetized, the left lungs were ligated, and the right lungs were gently lavaged three times via a tracheal tube with 0.7 mL phosphate-buffered saline. The total number of cells in the collected BALF was counted with a NucleoCounter (NC-250; ChemoMetec, Gydevang, Denmark). For differential cell counts, BALF cell smears were prepared using Cytospin (Thermo Fisher Scientific, Inc., Waltham, MA, USA) and stained with Diff-Quik solution (Dade Diagnostics, Aguada, Puerto Rico). The different cell types were counted (*n* = 200/slide). BALF was immediately centrifuged at 2000 rpm for 5 min, and the collected supernatant was stored at −70 °C until the cytokine levels were measured by enzyme-linked immunosorbent assay (ELISA).

### 4.3. Measurement of Cytokine Levels

IL-4, IL-5, and IL-13 levels in BALF were quantified by ELISA using commercial kits (Thermo Fisher Scientific) according to the manufacturer’s protocol.

### 4.4. Histological Analysis

Forty-eight hours after the final Kathon dose, mice were euthanized for histological examination. Lung tissue was removed, fixed in 10% (*v*/*v*) neutral-buffered formalin, dehydrated, embedded in paraffin, and cut into 4-μm sections. The sections were deparaffinized with xylene and stained with H&E, MT, and PAS (Sigma-Aldrich, St. Louis MO, USA). Stained sections were analyzed under a light microscope (Axio Imager M1; Carl Zeiss, Oberkochen, Germany). Each successive field was individually assessed to determine the severity of inflammatory cell infiltration, eosinophilic cell infiltration to perivascular/alveolar areas, mucus production, and goblet cell hyperplasia. The degree of lung injury was estimated by assigning a semi-quantitative score (0–4). The lesions were histologically graded by an experienced histopathologist using a blinded scoring system for the extent and severity of inflammation and fibrosis, as previously outlined [61].

### 4.5. RNA Isolation, Library Preparation, and Sequencing

Total RNA was isolated using Trizol reagent (Invitrogen, Carlsbad, CA, USA). RNA quality was assessed by an Agilent 2100 bioanalyzer using the RNA 6000 Nano Chip (Agilent Technologies, Amstelveen, Netherlands), and RNA quantification was performed using ND-2000 Spectrophotometer (Thermo Fisher Scientific). Only samples with an A260/A280 ratio >1.8 and RIN value >7 were considered suitable for use.

For control and test RNAs, a library was constructed using a QuantSeq 3′ mRNA-Seq Library Prep Kit (Lexogen, Inc., Wien, Austria) according to the manufacturer’s instructions. In brief, each 500-ng sample of total RNA was prepared, and an oligo-dT primer containing an Illumina-compatible sequence at its 5′ end was hybridized. Reverse transcription was then performed. After degradation of the RNA template, second-strand synthesis was initiated by a random primer containing an Illumina-compatible linker sequence at its 5′ end. The double-stranded library was purified by using magnetic beads to remove all reaction components. The library was amplified to add the complete adapter sequences required for cluster generation. The finished library was purified from PCR components. High-throughput sequencing was performed through single-end 75 sequencing using NextSeq 500 (Illumina, Inc., San Diego, CA, USA). The raw QuantSeq reads were trimmed using BBDuk (https://jgi.doe.gov/data-and-tools/bbtools/bb-tools-user-guide/bbduk-guide/) to remove adapter sequences, poly-A tails, and low-quality bases using following parameters; ref=polyA.fa.gz,truseq_rna.fa.gz k=13 int=f ktrim=r useshortkmers=t mink=5 qtrim=r trimq=10 minlength=20. Read quality before and after trimming was checked using FASTQC.

### 4.6. Data Analysis

QuantSeq 3′ mRNA-Seq reads were aligned using Bowtie2. Bowtie2 indices were either generated from genome assembly sequences or the representative transcript sequences for aligning to the genome or transcriptome. The alignment file was used for assembling transcripts, estimating their abundances, and detecting differential expression of genes. DEGs were determined based on counts from unique and multiple alignments using coverage in Bedtools (Quinlan AR, 2010). Read count (RC) data were processed based on the quantile normalization method using EdgeR within R and Bioconductor. Gene classification was based on searches done in the DAVID (http://david.abcc.ncifcrf.gov/) and Medline databases (http://www.ncbi.nlm.nih.gov/).

### 4.7. GO Category and Pathway Analysis

To classify the genes altered by Kathon exposure into groups with similar expression patterns, each gene was assigned to an appropriate category according to its main cellular function. To determine significantly over-represented GO findings, the DAVID functional annotation clustering tool was used by choosing the default option. Pathway analysis was performed to determine significant pathways for DEGs using microarray gene pathway annotations downloaded from KEGG (http://www.genome.jp/kegg/). Fisher’s exact test was used to find significant enrichment for pathways, and the resulting *p*-values were adjusted using the BH false discovery rate (FDR) algorithm. Pathway categories with FDR < 0.05 were reported. Additional pathways and disease-gene associations were analyzed using the CTD (http://ctdbase.org) and Ingenuity Pathway Analysis (IPA, Ingenuity Systems, Redwood City, CA, USA).

### 4.8. Statistical Analysis

All statistical analyses were performed using GraphPad Prism v.7 (GraphPad Software, San Diego, CA, USA). Statistical multiple comparisons were performed by one-way analysis of variance (ANOVA) followed by Dunnett and Bonferroni-adjusted Mann-Whitney tests. Data are expressed as the mean ± SD. Values were considered statistically significant at *p* < 0.05.

## 5. Conclusions

Taken together, the results of the current study suggest that repeated Kathon exposure may induce fibrotic inflammation resulting in a significant increase in fibrosis-related cytological and histological findings. Moreover, Kathon-induced fibrotic responses exhibited a unique pattern compared to PHMG-P induced fibrosis, with neutrophil and lymphocyte-associated features, which may be associated with eosinophil and Th2-associated fibrotic responses. These findings may help patients who complain of fibrosis are recognized as victims and help determine potential victims.

## Figures and Tables

**Figure 1 molecules-25-04684-f001:**
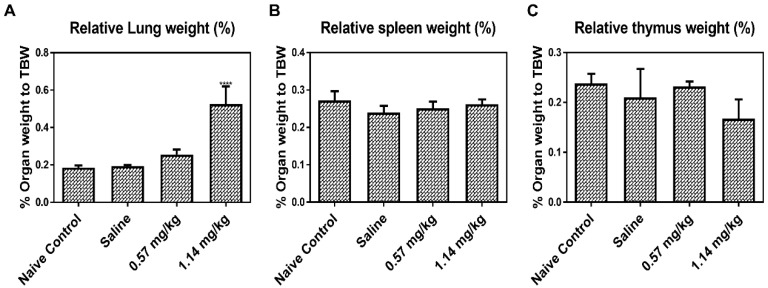
Changes in the relative lung, spleen, and thymus weights of normal control (NC), saline, and Kathon-exposed mice. Relative lung (**A**), spleen (**B**), and thymus (**C**) weights were calculated using the following formula: relative organ weight = organ weight (g)/terminal body weight (g) × 100%. Data are presented as the mean ±standard deviation (*n* = 5 per group). **** *p* < 0.0001 vs. saline.

**Figure 2 molecules-25-04684-f002:**
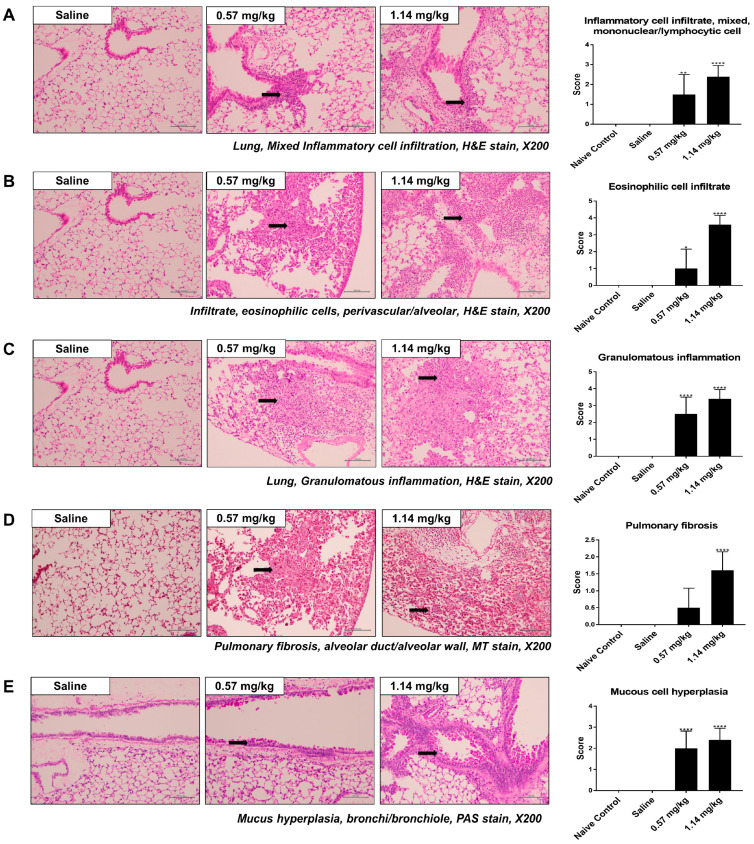
Histopathological analysis of lung tissues from control and Kathon-exposed mice. Representative hematoxylin and eosin (H&E)-stained lung sections and inflammation scores of mixed inflammatory cell infiltration (**A**), eosinophilic cell infiltration (**B**), and granulomatous inflammation (**C**). Masson’s trichrome (MT)-stained sections of lung and pulmonary fibrosis (**D**). Representative periodic acid–Schiff (PAS)-stained lung sections and mucus cell hyperplasia scores (**E**). Black arrows indicate inflammatory cells, collagen, and goblet cell deposition. * *p* < 0.05, ** *p* < 0.01, and **** *p* < 0.0001 vs. saline.

**Figure 3 molecules-25-04684-f003:**
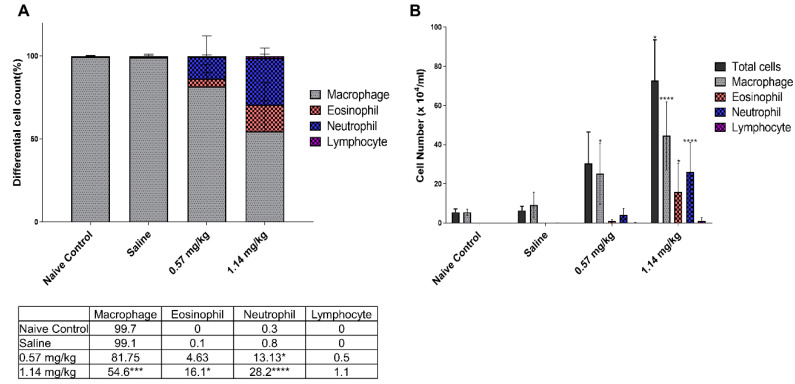
Effect of Kathon-induced changes on total and differential cell counts in the bronchoalveolar lavage fluid (BALF) of mice. Cell population composition as a percentage of total cells (**A**) and the number of total cells, macrophages, neutrophils, and lymphocytes (**B**) in the BALF of normal control (NC), saline, and Kathon mice. Data are presented as the mean ±standard deviation (*n* = 5 per group). * *p* < 0.05, *** *p* < 0.001, and **** *p* < 0.0001 vs. saline.

**Figure 4 molecules-25-04684-f004:**
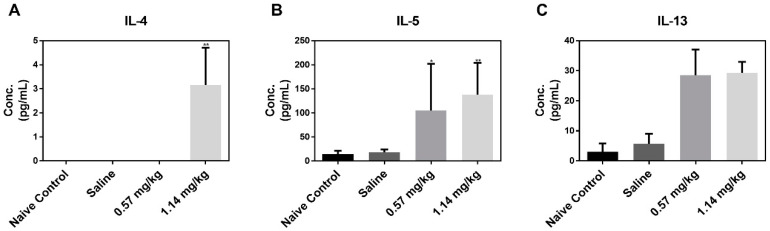
T helper 2 (Th2) cytokine levels in bronchoalveolar lavage fluid (BALF). Enzyme immunoassay of IL-4 (**A**), IL-5 (**B**), and IL-13 (**C**). Data are presented as the mean ±standard deviation (*n* = 5 per group). * *p* < 0.05 and ** *p* < 0.01 vs. saline.

**Figure 5 molecules-25-04684-f005:**
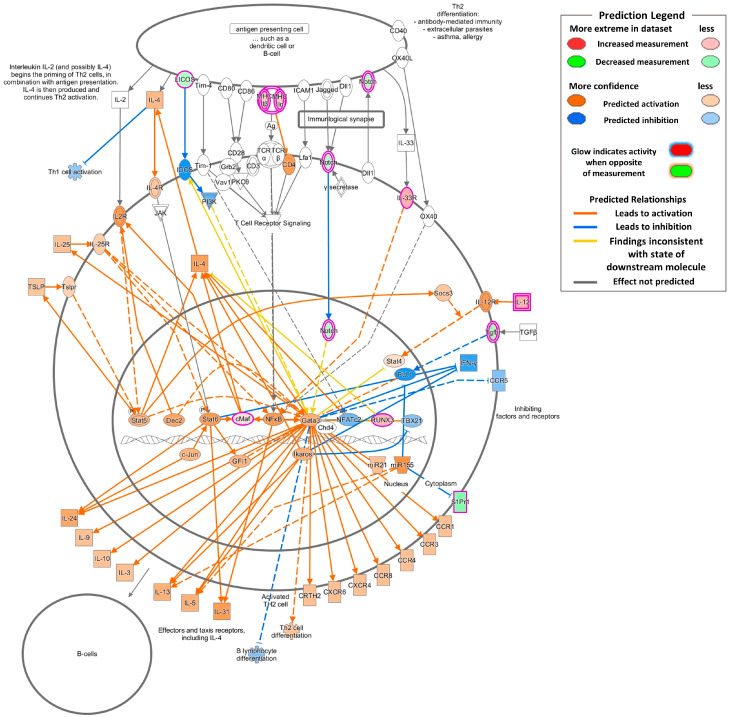
Kathon-activated T helper cell type-2 (Th2) pathways. Pathway image generated by IPA core analysis illustrating gene expression and potential outcomes associated with the Th2 pathway. Color intensity indicates the degree of upregulation (red) or downregulation (green) within the pathway and predicted activation (orange) or inhibition (blue). White indicates molecules present in the dataset but that did meet the cutoff values (false discovery rate ≤0.05 and a log2-fold change of either ≥1 or ≤−1).

**Table 1 molecules-25-04684-t001:** Gene ontology (GO) functional category analysis of Kathon-altered genes.

Term	Annotated Genes Quantity	*p* Value	Annotated Genes (*p* Value)
**Biological Processes**			
Inflammatory response	36	0.000	*GM21541*(0.01)*, RARRES2*(0.02)*, CYSLTR1*(0.01)*, PF4*(0.03)*, NLRP1A*(0.02)*, CX3CL1*(0.01)*, FPR2*(0.03)*, IL15*(0.03)*, TLR6*(0.02)*, TLR7*(0.01)*, CCRL2*(0.00)*, CHIL4*(0.00)*, HRH1*(0.01)*, REL*(0.02)*, ITGB6*(0.01)*, REG3G*(0.03)*, NOS2*(0.03)*, FAS*(0.01)*, HYAL1*(0.03)*, HC*(0.03)*, CCL21C*(0.03)*, EPHX2*(0.02)*, CCL21A*(0.01)*, AXL*(0.03)*, CCL21B*(0.02)*, CHST4*(0.01)*, GM13304*(0.01)*, GAL*(0.01)*, CHST1*(0.00)*, TNFRSF10B*(0.02)*, RELT*(0.03)*, GM10591*(0.01)*, PTGDR*(0.01)*, GM1987*(0.01)*, CAMK1D*(0.01)*, BMP6*(0.00)
Immune response	28	0.001	*GM21541*(0.01)*, PF4*(0.03)*, CX3CL1*(0.01)*, OAS2*(0.01)*, IL15*(0.03)*, TLR6*(0.02)*, TLR7*(0.01)*, CD74*(0.02)*, LIF*(0.04)*, FAS*(0.01)*, SMAD6*(0.01)*, CCL21C*(0.03)*, CTLA4*(0.01)*, CCL21B*(0.02)*, GM13304*(0.01)*, TNFSF8*(0.01)*, TNFRSF10B*(0.02)*, CD36*(0.02)*, CXCL14*(0.01)*, H2-EB2*(0.02)*, RELT*(0.03)*, PPBP*(0.01)*, GM10591*(0.01)*, H2-EB1*(0.01)*, TGFBR3*(0.04)*, H2-AA*(0.02)*, EDA*(0.00)*, BMP6*(0.00)
Cellular response to interleukin-1	17	0.000	*GM21541*(0.01)*, HYAL1*(0.03)*, HYAL2*(0.03)*, CCL21C*(0.03)*, MYLK3*(0.00)*, CCL21A*(0.01)*, CCL21B*(0.02)*, CX3CL1*(0.01)*, GM13304*(0.01)*, SOX9*(0.05)*, PCK1*(0.00)*, PTGIS*(0.02)*, GM10591*(0.01)*, ADAMTS12*(0.01)*, GM1987*(0.01)*, MYC*(0.05)*, FN1*(0.04)
Positive regulation of extracellular signal regulated kinase (ERK)1 and ERK2 cascade	17	0.043	*GM21541*(0.01)*, BMP4*(0.01)*, FGFR3*(0.04)*, CCL21C*(0.03)*, CCL21A*(0.01)*, CCL21B*(0.02)*, FGF10*(0.03)*, CX3CL1*(0.01)*, GM13304*(0.01)*, CD74*(0.02)*, KDR*(0.02)*, CD36*(0.02)*, GM10591*(0.01)*, VEGFA*(0.03)*, TEK*(0.01)*, ADRA1A*(0.03)*, GM1987*(0.01)
Positive regulation of mitogen-activated protein kinase (MAPK) cascade	15	0.001	*CAV2*(0.02)*, FGFR3*(0.04)*, FGF10*(0.03)*, IGF2*(0.04)*, PRKCE*(0.01)*, KDR*(0.02)*, LIF*(0.04)*, ADRB3*(0.03)*, DUSP19*(0.01)*, ADRB2*(0.00)*, CD36*(0.02)*, RELT*(0.03)*, BNIP2*(0.01)*, ADRA1A*(0.03)*, FAS*(0.01)
Cellular response to tumor necrosis factor	15	0.002	*GM21541*(0.01)*, HYAL1*(0.03)*, HYAL2*(0.03)*, CCL21C*(0.03)*, CCL21A*(0.01)*, CCL21B*(0.02)*, CX3CL1*(0.01)*, GM13304*(0.01)*, DCSTAMP*(0.02)*, PCK1*(0.00)*, GM10591*(0.01)*, SLC2A4*(0.00)*, COL1A1*(0.05)*, ADAMTS12*(0.01)*, GM1987*(0.01)
Notch signaling pathway	15	0.007	*HP*(0.03)*, KCNA5*(0.03)*, SOX9*(0.05)*, RCAN2*(0.05)*, HES1*(0.00)*, ASCL1*(0.04)*, PTP4A3*(0.00)*, HEY1*(0.00)*, DLL4*(0.00)*, MIB2*(0.04)*, PLN*(0.02)*, NOTCH4*(0.05)*, FOXC1*(0.00)*, TMEM100*(0.01)*, CFD*(0.01)
Extracellular matrix organization	15	0.003	*B4GALT1*(0.05)*, COL18A1*(0.01)*, RECK*(0.04)*, COL4A1*(0.04)*, EGFL6*(0.01)*, ELN*(0.05)*, SOX9*(0.05)*, NDNF*(0.03)*, LAMB3*(0.03)*, HPSE2*(0.05)*, CRISPLD2*(0.04)*, FOXF1*(0.01)*, COL6A4*(0.05)*, LAMC2*(0.05)*, FN1*(0.04)
Chemotaxis	14	0.010	*RARRES2*(0.02)*, CYSLTR1*(0.01)*, FGF10*(0.03)*, PF4*(0.03)*, FPR2*(0.03)*, ACKR4*(0.04)*, CX3CL1*(0.01)*, LSP1*(0.00)*, CCRL2*(0.00)*, CXCL17*(0.01)*, S1PR1*(0.01)*, ECSCR*(0.02)*, CX3CR1*(0.01)*, XCR1*(0.02)
Neutrophil chemotaxis	10	0.011	*GM21541*(0.01)*, ITGA9*(0.01)*, PPBP*(0.01)*, GM10591*(0.01)*, CCL21C*(0.03)*, CCL21A*(0.01)*, CCL21B*(0.02)*, GM13304*(0.01)*, CX3CL1*(0.01)*, GM1987*(0.01)
Cellular response to interferon-gamma	14	0.000	*WNT5A*(0.02)*, GM21541*(0.01)*, CCL21C*(0.03)*, CCL21A*(0.01)*, CCL21B*(0.02)*, GM13304*(0.01)*, CX3CL1*(0.01)*, DAPK3*(0.03)*, H2-Q7*(0.01)*, GM10591*(0.01)*, NOS2*(0.03)*, IL12B*(0.02)*, GM1987*(0.01)*, MYC*(0.05)
Chemokine-mediated signaling pathway	11	0.001	*GM21541*(0.01)*, CXCL17*(0.01)*, PPBP*(0.01)*, GM10591*(0.01)*, CCL21C*(0.03)*, CCL21A*(0.01)*, CCL21B*(0.02)*, PF4*(0.03)*, GM13304*(0.01)*, CX3CL1*(0.01)*, GM1987*(0.01)
Positive regulation of canonical Wnt signaling pathway	11	0.003	*WNT2*(0.03)*, CAV1*(0.01)*, FGFR3*(0.04)*, SULF2*(0.05)*, RSPO1*(0.02)*, FGF10*(0.03)*, SOX4*(0.02)*, COL1A1*(0.05)*, EDA*(0.00)*, DAPK3*(0.03)*, WNT2B*(0.03)
Transforming growth factor beta receptor signaling pathway	11	0.006	*ACVRL1*(0.01)*, SMAD9*(0.03)*, TGFBR1*(0.03)*, SMAD6*(0.01)*, COL3A1*(0.03)*, CLDN5*(0.02)*, COL1A2*(0.04)*, TGFBR3*(0.04)*, ENG*(0.01)*, HPGD*(0.00)*, CDH5*(0.03)
Collagen fibril organization	10	0.000	*FMOD*(0.03)*, P3H1*(0.02)*, TGFBR1*(0.03)*, COL3A1*(0.03)*, COL1A2*(0.04)*, FOXC1*(0.00)*, COL1A1*(0.05)*, LOXL2*(0.03)*, ADAMTS2*(0.03)*, COL5A1*(0.04)
**Kyoto Encyclopedia of Genes and Genomes (** **KEGG) Pathways**			
PI3K-Akt signaling pathway	35	0.000	*FGFR3*(0.04)*, TNC*(0.03)*, COL3A1*(0.03)*, FGF10*(0.03)*, GNG11*(0.05)*, BCL2L1*(0.03)*, LAMB3*(0.03)*, COL6A4*(0.05)*, TEK*(0.01)*, ITGB6*(0.01)*, COL6A3*(0.03)*, COL6A2*(0.02)*, TNN*(0.00)*, CREB3L3*(0.01)*, PRKAA2*(0.02)*, PPP2R2B*(0.05)*, MYC*(0.05)*, PPP2R2C*(0.00)*, AKT3*(0.01)*, FN1*(0.04)*, GHR*(0.02)*, COL4A1*(0.04)*, KITL*(0.02)*, COL5A1*(0.04)*, PCK1*(0.00)*, KDR*(0.02)*, VWF*(0.03)*, ITGA9*(0.01)*, CD19*(0.04)*, ITGA5*(0.00)*, VEGFA*(0.03)*, COL1A2*(0.04)*, LAMC2*(0.05)*, RELN*(0.04)*, COL1A1*(0.05)
Cytokine-cytokine receptor interaction	26	0.001	*GM21541*(0.01)*, TNFRSF12A*(0.05)*, BMPR2*(0.04)*, PF4*(0.03)*, IL15*(0.03)*, CX3CL1*(0.01)*, LIF*(0.04)*, FAS*(0.01)*, XCR1*(0.02)*, GHR*(0.02)*, TGFBR1*(0.03)*, CCL21C*(0.03)*, CCL21A*(0.01)*, EDA2R*(0.02)*, CCL21B*(0.02)*, GM13304*(0.01)*, TNFSF8*(0.01)*, TNFRSF10B*(0.02)*, CXCL14*(0.01)*, GM10591*(0.01)*, RELT*(0.03)*, PPBP*(0.01)*, CX3CR1*(0.01)*, IL12B*(0.02)*, EDA*(0.00)*, GM1987*(0.01)
Extracellular matrix (ECM)-receptor interaction	20	0.043	*COL4A1*(0.04)*, TNC*(0.03)*, COL3A1*(0.03)*, COL5A1*(0.04)*, GP9*(0.00)*, VWF*(0.03)*, ITGA9*(0.01)*, LAMB3*(0.03)*, CD36*(0.02)*, ITGA5*(0.00)*, COL6A4*(0.05)*, ITGB6*(0.01)*, COL6A3*(0.03)*, COL6A2*(0.02)*, COL1A2*(0.04)*, RELN*(0.04)*, TNN*(0.00)*, LAMC2*(0.05)*, COL1A1*(0.05)*, FN1*(0.04)
Calcium signaling pathway	17	0.001	*CCKAR*(0.04)*, ADCY4*(0.05)*, ADORA2B*(0.01)*, CYSLTR1*(0.01)*, MYLK3*(0.00)*, MYLK4*(0.00)*, EDNRB*(0.01)*, ADRB3*(0.03)*, ATP2B2*(0.03)*, HRH1*(0.01)*, ADRB2*(0.00)*, PDE1B*(0.03)*, PLN*(0.02)*, PLCG2*(0.03)*, AVPR1A*(0.03)*, ADRA1A*(0.03)*, NOS2*(0.03)
Metabolism of xenobiotics by cytochrome P450	13	0.000	*GSTA3*(0.00)*, CYP2F2*(0.02)*, ALDH3B2*(0.01)*, EPHX1*(0.05)*, DHDH*(0.00)*, MGST3*(0.01)*, GSTM1*(0.00)*, GSTM2*(0.01)*, GSTM3*(0.03)*, GSTK1*(0.02)*, HSD11B1*(0.02)*, GSTP2*(0.01)*, GSTP1*(0.00)
Peroxisome proliferator-activated receptors (PPAR) signaling pathway	10	0.002	*1700061G19RIK*(0.04)*, ACSL1*(0.03)*, CD36*(0.02)*, SORBS1*(0.02)*, FABP3*(0.01)*, FABP1*(0.00)*, AQP7*(0.05)*, FABP7*(0.01)*, ADIPOQ*(0.01)*, PCK1*(0.01)

Statistically significant GO terms were only considered (*p* value ≤ 0.05). Certain genes are counted in more than one annotation category.

**Table 2 molecules-25-04684-t002:** Comparative Toxicogenomics Database (CTD) disease category analysis of genes altered by Kathon exposure.

Disease Name	Annotated Genes Quantity	*p* Value	Annotated Genes(*p* Value)
Respiratory tract diseases	107	2.36 × 10^−32^	*ACE2*(0.01)*, ACTA2*(0.04)*, ACVRL1*(0.01)*, ADAMTS2*(0.03)*, ADCYAP1R1*(0.01)*, ADGRE5*(0.01)*, ADIPOQ*(0.01)*, ADRB2*(0.00)*, ALDH2*(0.01)*, AREG*(0.04)*, ASCL1*(0.04)*, AXL*(0.03)*, BCHE*(0.00)*, BCL2L1*(0.03)*, BMPR2*(0.04)*, BTK*(0.04)*, CAT*(0.01)*, CAV1*(0.01)*, CCDC40*(0.03)*, CD74*(0.02)*, CDH13*(0.01)*, CDH23*(0.00)*, CEACAM1*(0.02)*, CFD*(0.01)*, CLDN5*(0.02)*, COL3A1*(0.03)*, CPE*(0.00)*, CRISPLD2*(0.04)*, CTLA4*(0.01)*, CX3CL1*(0.01)*, CXCL14*(0.01)*, CYB5A*(0.00)*, CYSLTR1*(0.01)*, DNAAF3*(0.05)*, DNAAF5*(0.02)*, DNAH5*(0.05)*, DNMT3A*(0.02)*, EDNRB*(0.01)*, EFEMP1*(0.03)*, EFNB2*(0.03)*, ELN*(0.05)*, ENG*(0.01)*, EPHX1*(0.05)*, FAS*(0.01)*, FGD6*(0.01)*, FN1*(0.04)*, FOXF1*(0.01)*, GJA1*(0.02)*, GPX3*(0.02)*, GSTM1*(0.00)*, GSTM2*(0.01)*, GSTP1*(0.00)*, HES1*(0.00)*, HEY1*(0.00)*, HILPDA*(0.00)*, HRH1*(0.01)*, ICAM2*(0.01)*, IER2*(0.03)*, IGFBP6*(0.00)*, IL12B*(0.02)*, IL15*(0.03)*, IL1RL1*(0.02)*, ITGB6*(0.01)*, KCNA5*(0.03)*, LAMC2*(0.05)*, LYST*(0.01)*, MAPT*(0.01)*, MERTK*(0.05)*, MMP10*(0.04)*, MSLN*(0.01)*, MYC*(0.05)*, MYCL*(0.01)*, NEK2*(0.01)*, NNAT*(0.00)*, NOS2*(0.03)*, NQO1*(0.01)*, OAS2*(0.01)*, PF4*(0.03)*, PLLP*(0.01)*, PON1*(0.01)*, PPBP*(0.01)*, PRDX6*(0.01)*, PTGDR*(0.01)*, PTGIS*(0.02)*, RAMP2*(0.02)*, RECK*(0.04)*, RIMS2*(0.01)*, RUNX3*(0.05)*, SCGB1A1*(0.03)*, SCNN1B*(0.01)*, SELENBP1*(0.00)*, SLC34A2*(0.02)*, SLC6A4*(0.01)*, SMAD9*(0.03)*, SMARCAL1*(0.01)*, SOD1*(0.01)*, SOX9*(0.05)*, TFF1*(0.05)*, THBD*(0.05)*, TMPRSS4*(0.05)*, TNC*(0.03)*, TNFSF8*(0.01)*, TNNT2*(0.03)*, UBE2L6*(0.03)*, UCHL1*(0.05)*, VEGFA*(0.03)*, WNT5A*(0.02)
Lung diseases, obstructive	25	1.52 × 10^−12^	*ADCYAP1R1*(0.01)*, ADRB2*(0.00)*, ALDH2*(0.01)*, AREG*(0.04)*, CAT*(0.01)*, CTLA4*(0.01)*, CXCL14*(0.01)*, DNAH5*(0.05)*, ELN*(0.05)*, EPHX1*(0.05)*, FOXF1*(0.01)*, GSTM1*(0.00)*, GSTP1*(0.00)*, IL1RL1*(0.02)*, ITGB6*(0.01)*, MMP10*(0.04)*, NOS2*(0.03)*, NQO1*(0.01)*, PTGDR*(0.01)*, SCNN1B*(0.01)*, SOD1*(0.01)*, TNC*(0.03)*, TNFSF8*(0.01)*, TNNT2*(0.03)*, VEGFA*(0.03)
Respiratory hypersensitivity	22	7.48 × 10^−10^	*ADCYAP1R1*(0.01)*, ADRB2*(0.00)*, ALDH2*(0.01)*, AREG*(0.04)*, CAT*(0.01)*, CLDN5*(0.02)*, CTLA4*(0.01)*, CXCL14*(0.01)*, CYSLTR1*(0.01)*, DNAH5*(0.05)*, GSTM1*(0.00)*, GSTP1*(0.00)*, HRH1*(0.01)*, IL15*(0.03)*, IL1RL1*(0.02)*, MMP10*(0.04)*, NOS2*(0.03)*, NQO1*(0.01)*, PTGDR*(0.01)*, SOD1*(0.01)*, TNC*(0.03)*, VEGFA*(0.03)
Carcinoma, bronchogenic	20	2.03 × 10^−5^	*AREG*(0.04)*, ASCL1*(0.04)*, AXL*(0.03)*, CAT*(0.01)*, CD74*(0.02)*, CDH13*(0.01)*, GSTM2*(0.01)*, GSTP1*(0.00)*, HES1*(0.00)*, LAMC2*(0.05)*, MYC*(0.01)*, MYCL*(0.04)*, NEK2, NNAT*(0.00)*, NQO1*(0.01)*, RECK*(0.04)*, RIMS2*(0.01)*, SLC34A2*(0.02)*, UCHL1*(0.05)*, VEGFA*(0.03)
Asthma	19	1.75 × 10^−8^	*ADCYAP1R1*(0.01)*, ADRB2*(0.00)*, ALDH2*(0.01)*, AREG*(0.04)*, CAT*(0.01)*, CTLA4*(0.01)*, CXCL14*(0.01)*, CYSLTR1*(0.01)*, DNAH5*(0.05)*, GSTM1*(0.00)*, GSTP1*(0.00)*, IL1RL1*(0.02)*, MMP10*(0.04)*, NOS2*(0.03)*, NQO1*(0.01)*, PTGDR*(0.01)*, SOD1*(0.01)*, TNC*(0.03)*, VEGFA*(0.03)
Pulmonary fibrosis	12	0.01161	*ACE2*(0.01)*, ACTA2*(0.04)*, ADIPOQ*(0.01)*, AREG*(0.04)*, CAT*(0.01)*, CFD*(0.01)*, COL3A1*(0.03)*, ELN*(0.05)*, FN1*(0.04)*, IL12B*(0.02)*, SOD1*(0.01)*, WNT5A*(0.02)
Pulmonary disease, chronic obstructive	9	0.01999	*ELN*(0.05)*, EPHX1*(0.05)*, ITGB6*(0.01)*, NOS2*(0.03)*, SCNN1B*(0.01)*, TNFSF8*(0.01)*, TNNT2*(0.03)*, VEGFA*(0.03)

**Table 3 molecules-25-04684-t003:** Genes altered in the Th2-related signaling pathways.

Th2 Pathway			
Symbol	Entrez Gene Name	Fold Change	*p* Value
BMPR2	bone morphogenetic protein receptor, type II (serine/threonine kinase)	0.62	0.04
H2-Eb2	histocompatibility 2, class II antigen E beta2	2.95	0.02
H2-Q7	major histocompatibility complex, class I, A	2.10	0.01
H2-Aa	major histocompatibility complex, class II, DQ alpha 1	2.23	0.02
H2-Eb1	major histocompatibility complex, class II, DR beta 5	2.52	0.01
Icosl	inducible T cell costimulator ligand	0.62	0.04
IL12B	interleukin 12b	3.50	0.02
IL1RL1	interleukin 1 receptor-like 1	3.46	0.02
MAF	avian musculoaponeurotic fibrosarcoma oncogene homolog	1.71	0.05
NOTCH4	notch 4	0.62	0.05
RUNX3	runt related transcription factor 3	3.02	0.05
S1PR1	sphingosine-1-phosphate receptor 1	0.50	0.01
TGFBR1	transforming growth factor, beta receptor I	2.09	0.03
TGFBR3	transforming growth factor, beta receptor III	0.51	0.04
**IL-4 Signaling**			
**Symbol**	**Entrez Gene Name**	**Fold Change**	***p* Value**
AKT3	thymoma viral proto-oncogene 3	0.59	0.01
H2-Eb2	histocompatibility 2, class II antigen E beta2	2.95	0.02
H2-Q7	major histocompatibility complex, class I, A	2.10	0.01
H2-Aa	major histocompatibility complex, class II, DQ alpha 1	2.23	0.02
H2-Eb1	major histocompatibility complex, class II, DR beta 5	2.52	0.01

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
