# Peer review of "Kathon Induces Fibrotic Inflammation in Lungs: The First Animal Study Revealing a Causal Relationship between Humidifier Disinfectant Exposure and Eosinophil and Th2-Mediated Fibrosis Induction"

_molecules, 2020, doi:10.3390/molecules25204684_

Round 1

Reviewer 1 Report

The manuscript entitled :"Kathon induces fibrotic inflammation in lungs: The first animal study revealing the causal relationship between humidifier disinfectant exposure and eosinophil and Th2-mediated fibrosis induction" focused on the evaluation of humidifer disinfectant in the development of lung fibrosis requirese several major changes to be suitable for publication.

  • In the introduction section, the authors partially showed the worldwide relevance of this phenomenon, in my opinion they should better underlyne the clinical problem derivd from this issue. An extensive revision literatur data is required.
  • According to the last issue, the authors should well define the population cohort that may be affected from this issue.
  • In the experimental section, several major issue were identified. In my opinion, the authors well investigate the in vivo experimental section related to the analysis embraced on murine models. This analysis is not adequately performed on cell culture section. My suggestion is to better perform molecular analysis by evaluating the EC50, and relative effects on cell cultures. In addition, cell counting, vitality and damage should be reported by the authors.
  • In the material and method section, the authors report  technical features of RNA-seq performed on experimental samples. According to this issue, a relevant aspect is related to the clarification of experimental procedure approached to extract RNA. In my opinion, RNA quantification data should be reported in a supplementary table. In addition, could they report which were check paramters of NGS run that the authors evaluated?
  • In the experimental section, the authors define the infusion methodology to transfer Kathon in murine models. This aspect may represent a technical issue because this modality is different respect the conventional exposition modality for this molecule. Please, could the authors discuss this limitation?
  • In the table 1 the authors report p-value of each group of analyzed genes. In my opinion, the authors should define p-value for each of candidate genes evaluated in this section in order to confirm if each of them is differentially expressed.

Author Response

Response to Reviews

Thank you very much for your detailed and thoughtful comments. The followings are the summary of the revisions.   

# Reviewer: 1

General comments to authors

The manuscript entitled "Kathon induces fibrotic inflammation in lungs: The first animal study revealing a causal relationship between humidifier disinfectant exposure and eosinophil and Th2-mediated fibrosis induction" focused on the evaluation of humidifier disinfectant in the development of lung fibrosis requires several major changes to be suitable for publication.

* Comments;

1) In the introduction section, the authors partially showed the worldwide relevance of this phenomenon, in my opinion they should better underlying the clinical problem derived from this issue. An extensive revision literature data is required. According to the last issue, the authors should well define the population cohort that may be affected from this issue.

Answer:

We added the clinical evidence linking humidifier disinfectant-associated lung injury (HDLI) and exposure to humidifier disinfectants (HDs) containing 5-chloro-2-methyl-4-isothiazolin-3-one (CMIT)/2-methyl-4-isothiazolin-3-one (MIT) in introduction section (page 2, line 48-53).

2) In the experimental section, several major issue were identified. In my opinion, the authors well investigate the in vivo experimental section related to the analysis embraced on murine models. This analysis is not adequately performed on cell culture section. My suggestion is to better perform molecular analysis by evaluating the EC50, and relative effects on cell cultures. In addition, cell counting, vitality and damage should be reported by the authors.

Answer:

We already identified the pulmonary toxic effects using in vitro cell system. A recent study in our research group showed that Kathon induced apoptotic cell death along with membrane damage in human bronchial epithelial (BEAS-2B) cells (Environ. Toxicol. 2020;35:27–36). We thought that pulmonary epithelial cell damage caused by Kathon is related to direct damage in lung tissue, and a single ITI dosing experiment(PHMG-P induced fibrosis was observed in single ITI dosing schedule) using ICR mice was performed. On day 14 after a single instillation with Kathon, several inflammatory responses such as the total number of BALF cells and the levels of TNF-α, IL-5, IL-13, MIP-1α, and MCP-1α clearly increased in the lung of mice. The proportion of natural killer cells and eosinophils were also significantly elevated in the spleen and the bloodstream, respectively. Therefore, from these results, we suggest that Kathon may induce eosinophilia-mediated disease in the lung by disrupting homeostasis of pulmonary surfactants. Considering that eosinophilia is closely related to fibrosis, current studies with another mice strain(C57BL/6, one of the most widely used mouse strains for chemical-induced fibrosis) and repeat ITI dosing schedule are performed to confirm and understand the relationship between them. And detailed mechanism studies using in vitro system are currently in progress in our research group. We hope this explanation and our recent study help you to understand and if my explanation doesn’t make sense to you, please contact me with any questions you have. We added these explanations in discussion section (page 10, line 159-171).

3) In the material and method section, the authors report technical features of RNA-seq performed on experimental samples. According to this issue, a relevant aspect is related to the clarification of experimental procedure approached to extract RNA. In my opinion, RNA quantification data should be reported in a supplementary table. In addition, could they report which were check parameters of NGS run that the authors evaluated?

Answer:

We mentioned the condition for RNA QC in M&M section, 4.5. RNA isolation, library preparation, and sequencing part (page 14, line 367-368), and provided QC results in result section (Table S2).

Also, the raw QuantSeq reads were trimmed using BBDuk (https://jgi.doe.gov/data-and-tools/bbtools/bb-tools-user-guide/bbduk-guide/) to remove adapter sequences, poly-A tails, and low-quality bases using following parameters; ref=polyA.fa.gz,truseq_rna.fa.gz k=13 int=f ktrim=r useshortkmers=t mink=5 qtrim=r trimq=10 minlength=20. Read quality before and after trimming was checked using FASTQC. We added these information in M&M section, 4.5. RNA isolation, library preparation, and sequencing part (page 14, line 378-382).

4) In the experimental section, the authors define the infusion methodology to transfer Kathon in murine models. This aspect may represent a technical issue because this modality is different respect the conventional exposition modality for this molecule. Please, could the authors discuss this limitation?

Answer:

We agree that it is necessary to evaluate the toxicity of each HD components include Kathon in inhalation exposure studies. However, considering the cost, quantity of test materials, and specialized techniques required for inhalation studies and the low number of available facilities where inhalation studies can be performed, it is not practical to conduct inhalation exposure studies. Also, in our previous study to investigate the PHMG-P-induced fibrotic effects, we confirmed and validated that intratracheally instilled animals exhibited similar fibrotic features with inhaled animals. Therefore, we propose using intratracheal instillation to identify HD-induced toxic effects that require further testing using inhalation studies. In current study, we confirmed the Kathon induced-fibrotic responses in intratracheally instilled mice. Further studies on inhalation exposure are needed to validate the fibrosis-inducing effects, and we plan to perform the inhalation study. We mentioned these information in discussion section (page 13, line 303-309).

5) In the table 1 the authors report p-value of each group of analyzed genes. In my opinion, the authors should define p-value for each of candidate genes evaluated in this section in order to confirm if each of them is differentially expressed.

Answer:

We added the p-value for each of candidate genes in table 1, table 2, and table 3 as reviewer’s comment.

Reviewer 2 Report

In this research study, the authors investigated whether a HD containing CMIT and MIT induces fibrotic lung injury following direct lung exposure in an animal model.  Different triggers may initiate injury and contribute to the fibrotic process in the lungs. However, there is currently a lack of evidence validating and identifying the causal relationship between HDs containing CMIT and/or MIT and fibrotic lung injury. The study design seems appropriate and the data are convincing. Limitations of the study are recognized by the authors. I believe the paper would be improved by the following minor revisions:

  1. General

The English language of the manuscript requires moderate improvement. The text would benefit from revision by the native speaker or English language medical writing corrections.

The manuscript is lacking separate sections: Introduction, Results, Discussion, Materials and Methods, Conclusions as requested by the Authors guidelines for publications in the Molecules Journal. The authors combined the Results and Discussion section into one. This may be confusing for some readers and not acceptable by the Journal publication policy. 

Author Response

Response to Reviews

Thank you very much for your detailed and thoughtful comments. The followings are the summary of the revisions.   

# Reviewer: 2

General comments to authors

In this research study, the authors investigated whether a HD containing CMIT and MIT induces fibrotic lung injury following direct lung exposure in an animal model. Different triggers may initiate injury and contribute to the fibrotic process in the lungs. However, there is currently a lack of evidence validating and identifying the causal relationship between HDs containing CMIT and/or MIT and fibrotic lung injury. The study design seems appropriate and the data are convincing. Limitations of the study are recognized by the authors. I believe the paper would be improved by the following minor revisions:

* Comments;

1) The English language of the manuscript requires moderate improvement. The text would benefit from revision by the native speaker or English language medical writing corrections.

Answer:

Actually, the English in this manuscript has been already checked by at least two professional editors, both native speakers of English. However, as reviewer’s comment, overall manuscript edited again by an editing service.

If the English in this manuscript doesn’t make sense to you, or if reviewer would like to point out grammar of my manuscript in detail, I will correct again that with pleasure.

2) The manuscript is lacking separate sections: Introduction, Results, Discussion, Materials and Methods, Conclusions as requested by the Authors guidelines for publications in the Molecules Journal. The authors combined the Results and Discussion section into one. This may be confusing for some readers and not acceptable by the Journal publication policy.

Answer:

We separated the Results and Discussion sections and added the Conclusion section into the manuscript as reviewer’s comment.

# Additional changes

1) According to the editing text contents, we added the reference and changed the reference #.

2) According to the editing text contents, section # was changed.

3) According to reviewer’s comment, overall manuscript edited again by an editing service.

Round 2

Reviewer 1 Report

The manuscript is now suitable for publication in the present form